# Can Milk Affect Recovery from Simulated Team-Sport Match Play?

**DOI:** 10.3390/nu12010112

**Published:** 2019-12-31

**Authors:** Paula Rankin, Danielle Callanan, Kevin O’Brien, Gareth Davison, Emma J. Stevenson, Emma Cockburn

**Affiliations:** 1Department of Science and Health, Institute of Technology Carlow, R93 V960 Carlow, Ireland; danielle.callanan@itcarlow.ie; 2School of Biomedical, Nutritional and Sport Sciences, Newcastle University, Newcastle Upon Tyne NE2 4HH, UK; emma.cockburn@ncl.ac.uk; 3School of Health and Human performance, Dublin City University, D09 V209 Dublin, Ireland; kevin.obrien@dcu.ie; 4Sport and Exercise Sciences Research Institute, Ulster University, Jordanstown BT37 OQB, UK; gw.davison@ulster.ac.uk; 5Population Health Sciences Institute, Newcastle University, Newcastle Upon Tyne NE2 4HH, UK; emma.stevenson@ncl.ac.uk

**Keywords:** performance, protein, muscle damage, post-exercise recovery, female

## Abstract

This study investigated the effects of cow’s milk on recovery from repeated simulated team games (STGs) in females. Twenty female team-sport athletes completed an STG circuit (2x ~ 30 min, with 10 min ‘half-time’). Measures of muscle function, soreness and tiredness, symptoms of stress and serum markers of muscle damage and oxidative stress were determined pre- and 24 h, 48 h, 72 h and 96 h following the circuit. At 48 h, a second STG was completed. Sprint performance (5 m, 15 m), lap time, countermovement jump (CMJ), heart rate and RPE were recorded during each STG. Immediately following each STG, participants consumed either 500 mL of cow’s milk (MILK) or 500 mL of an energy-matched carbohydrate (CHO) solution. Compared to CHO, MILK had a beneficial effect in attenuating losses in peak torque for knee extension (60°/s) (likely; effect size (ES) = 0.26 to 0.28) knee flexion (60°/s) (likely; ES = 0.45 to 0.61). A benefit for MILK was observed for 5 m sprint (possible-likely; ES = 0.40 to 0.58), 10 m sprint (likely; ES = 0.30 to 0.53) and symptoms of stress (likely–very likely, small). Mostly unclear outcomes for other variables were observed. For STG variables, trivial (HR, CMJ) and unclear (5 m sprint, 15 m sprint, lap-time, RPE) outcomes were recorded. In conclusion, the consumption of 500 mL of milk attenuated losses in muscle function and perceptions of stress following repeated simulated team-sports games. However, further investigation is warranted to determine whether MILK can influence subsequent team-sport performance.

## 1. Introduction

Participation in a single team-sport match results in fatigue, muscle damage and oxidative stress in males [1,2] and females [3,4]. In reality, team-sport athletes engage in multiple training sessions or games within short periods of time resulting in insufficient recovery between bouts [5,6,7]. Recently, Mohr et al. [8] investigated responses to three soccer games played within one week, with male participants. Results indicated that the largest physiological stress, fatigue and decreased performance followed the second game which was preceded by only 3 days recovery. Based on this literature, opportunities exist to explore the responses of female team-sport athletes to repeated match play with short recovery duration.

Minimizing muscle damage and enhancing recovery between training sessions or games could be beneficial for maintaining performance in subsequent bouts. Recent research has shown that the consumption of cow’s milk following muscle damaging exercise can attenuate decreases in muscle functional capacity in males [9,10] and females [11,12], perhaps due to an increase in post-exercise protein synthesis or a reduction in post-exercise degradation of muscle proteins [13]. It has also been reported that cow’s milk has anti-oxidant properties that may reduce oxidative stress [14] and the consumption of carbohydrate-protein mixtures may reduce lipid peroxidation [15]. The lack of effect of milk on exercise recovery reported in a number of investigations may be due to aspects of experimental design such as beverage blinding, homogeneity of participants, or the amount or timing of the cow’s milk intake [16]. To date, it is unknown whether the consumption of cow’s milk can enhance recovery from repeated match play, or whether a positive effect on recovery markers can influence subsequent match performance.

Given the benefits of cow’s milk for enhanced recovery following muscle damaging protocols, and with the short recovery periods experienced by athletes between team-sport training sessions and games, it makes the expectation tenable that post-exercise consumption of cow’s milk may combat the negative effects of muscle damage and fatigue, thus accelerating the rate of recovery between matches. Accordingly, this study aims to examine the effect of cow’s milk on recovery from a simulated team game and a second simulated team game following a 48 h recovery period. Additionally, the effect of cow’s milk on ‘in-game’ measures in the second simulated game is explored.

## 2. Materials and Methods

### 2.1. Participants

Twenty female team-sport athletes (age 20.6 ± 3.0 year; height 164.7 ± 5.0 cm; mass 65.4 ± 9.2 kg), from the amateur sports of camogie and ladies Gaelic football, participated in this investigation. All participants were training for and competing in their chosen sport at the time of data collection. Eligibility was assessed with a screening questionnaire, and volunteers were excluded from the study if they met any of the following criteria: current or recent use of nutritional supplements, intolerance to dairy or lactose products, injury in previous 3 months, surgery in previous 6 months, known coronary disease, uncontrolled metabolic disorder or respiratory disease, pregnancy and/or post-partum. Following verbal and written briefings written informed consent was provided. Ethics approval was gained from the Ethics Committees of Middlesex University and Institute of Technology Carlow, where data collection took place.

### 2.2. Study Design

Participants visited the laboratory on seven occasions. The first visit was comprised of study briefing and familiarization with all variables. Baseline values for all measures were determined at the second visit, following which participants were divided into two groups, cow’s milk (MILK) and an energy-matched carbohydrate (CHO) solution, matched according to player position and peak torque of the dominant leg knee extensors (60°/s). Data collection took place at the same time each day following 24 h abstinence from alcohol, caffeine and exercise.

Participants completed all measures following an overnight fast and a standardized snack. No more than 5 days post-baseline measures, participants completed an exercise protocol comprising a circuit with simulated team-sport games (STG1). Participants returned to the laboratory at 24 h and 48 h post-exercise to repeat baseline measures. Upon completion of measures 48 h post-exercise, participants repeated the circuit (STG2) returning again to repeat baseline measures 72 h and 96 h post STG1. Participants were asked to refrain from exercise for the duration of the experimental period, and from treating symptoms of muscle damage.

### 2.3. Exercise Protocol

Participants completed a modified version of a validated protocol, described in [17], designed to simulate team-sport exercise. Briefly, following a warm-up, participants completed 2x ~ 30 min halves of intermittent running in a circuit layout, separated by a 10 min half-time to simulate the format of camogie and ladies Gaelic football games. The circuit took ~51–53 s to complete. Each lap commenced every 80 s, and 22 laps were completed in each half. The circuit was ~125 m in length and included three maximal sprints (2 × 10 m, 1 × 20 m), striding (27 m), an agility exercise (12 m) and countermovement jumps (2x), all separated by walking (20 m) and jogging (26 m). Sprint performance, countermovement jump height (CMJ), lap time, heart rate (HR) and rate of perceived exertion (RPE) were recorded for each lap of the circuit. Upon completion of the STG, participants completed a cool-down, immediately after which they consumed their assigned drink.

### 2.4. Nutritional Intervention and Dietary Control

Thirty minutes prior to commencing each STG, each participant consumed a snack (Oats and Honey cereal bar, Nature Valley, Middlesex, UK). Immediately following the STG, participants were provided with either 500 mL of cow’s milk (Avonmore 1% Light, Glanbia, Ireland) or 500 mL of an energy-matched carbohydrate solution, which was consumed within 30 min. Participants were not blinded to the beverage received. Macronutrient composition (per 500 mL) of the cow’s milk was as follows: Energy 910 kJ/215 kcal, Protein 17.0 g, Carbohydrate 25.5 g, and Fat 5.0 g. The energy-matched carbohydrate solution (Energy 910 kJ/215 kcal, Protein 0.2 g, Carbohydrate 53.6 g, Fat 0.0 g) consisted of glucose mixed with water and a flavoured cordial. Each participant completed a food diary for 24 h prior to testing and for the subsequent 4 days. Participants were provided with a weighing scale and measuring jug and were instructed to follow their usual eating habits. Analysis (Nutritics Professional Diet Analysis, Nutritics Ltd., Dublin, Ireland) indicated that there were no differences in energy, carbohydrate, protein or fat intake between groups.

### 2.5. Outcome Measures

#### 2.5.1. Blood Sampling and Biochemical Analysis

Each day, a blood sample was collected by venepuncture from a forearm vein in serum separator (SST) tubes. A sample was also collected 2 h post-completion of the STG. Samples were centrifuged, aliqouted and stored at −70 °C for later analysis. Serum creatine kinase (CK) and high sensitivity C-reactive protein (hsCRP) were determined singly using high-sensitivity procedures (Roche Cobas 6000 chemistry module c501, Hoffmann-La Roche, Basel, Switzerland). Lipid hydroperoxides (LOOHs) were determined singly using the ferrous oxidation of xylenol orange method (FOX 1), using a modification of previous methods [18,19]. Absorbance was read spectrophotometrically (U-2001, Hitachi, England) at 560 nm against a linear standard curve (range 0–5 μmol·L^−1^). Protein Carbonyls (PCs) were determined using a commercially available kit (Abcam, Cambridge, UK) utilising a methodology based on that described by Levine et al. [20].

#### 2.5.2. Muscle Soreness and Muscle Tiredness

Muscle soreness during the completion of a squat to approximately 90° knee flexion, and during isokinetic testing, was measured on a visual analogue scale (VAS), with subjects rating soreness on a scale of 0 (no soreness) to 10 (as bad as it could be). A similar VAS was used to measure muscle tiredness.

#### 2.5.3. Symptoms of Stress

Symptoms of stress were monitored using the Daily Analysis of Life Demands (DALDA) questionnaire [21]. The questionnaire is comprised of two parts: Part A represents the sources of stress, and Part B assesses the manifestation of this stress in the form of symptoms. Participants marked each statement as being either ‘normal’, ‘worse than normal’, or ‘better than normal’. Data for Part B is presented.

#### 2.5.4. Muscle Function

To determine peak torque (Nm), participants completed three dominant-leg maximal effort knee extension and flexion repetitions at 60°/s and 180°/s, with 60 s recovery between speeds, on a Biodex System3 Isokinetic dynamometer (Biodex Medical System, Shirley, NY, USA). Rate of force development (RFD) was determined over 100–200 ms of an isometric contraction [22]. Briefly, two maximal 5 s isometric contractions of the quadriceps were executed with the knee fixed at 70°. RFD was calculated over the time interval of 100–200 (Δtorque/Δtime) relative to the onset of contraction, which was defined as the time point when torque generated exceeded the baseline by >7.5 Nm [23]. CMJ height was determined using an Optojump optical measurement system (Microgate, Bolzano, Italy). Participants completed three trials employing standard countermovement jump technique, and jump height was calculated from flight time. The highest recorded jump was used for analysis. Reactive strength index (RSI) was determined by dividing jump height (cm) by ground contact time (cm) following three maximal effort drop jumps from a height of 45 cm. Hands were maintained on the hips during the jump to eliminate any contribution of arm swing. Ground contact time and jump height were measured using the Optojump optical measurement system and the greatest RSI was used for analysis. Twenty metre sprint performance, with split 5 m and 10 m times, from a standing start 20 cm behind the start line, was recorded using timing gates (Microgate Racetime 2, Bolzano, Italy). Participants completed three sprints with a rest time of 120 s between sprints with the best time used for analysis.

### 2.6. Data Analysis

Data analysis was carried out by making probabilistic magnitude-based inferences about the true values of the effect of intervention on outcomes by expressing the uncertainty as 90% confidence limits (CLs) [24]. Within-group effects over time and the effect of MILK versus CHO were determined using a published spreadsheet [25]. The magnitude of the smallest substantive effect for muscle function variables was determined as the smallest standardised (Cohen) difference in the mean (0.2 times the between-subjects standard deviation for all participants). The smallest worthwhile effect for muscle soreness, tiredness and DALDA was set at 10% of the range of the scale [26]. Comparisons were made between baseline and subsequent timepoints. To overcome heteroscedastic error, data for peak torque, RFD, countermovement jump, RSI, sprint performance, lap times and HR were log-transformed [27]. Soreness, tiredness, DALDA and RPE values were not log-transformed because of interval scaling [27]. Because of large percentage changes, serum markers are reported as factors [28]. Means of log-transformed data were subsequently back transformed to provide mean percentage change and percentage SD. Chances of change over time and of benefit and harm were assessed qualitatively as follows: <1% almost certainly none, 1–5% very unlikely, 5–25% unlikely, 25–75% possibly, 75–95% likely, 95–99% very likely, >99% almost certainly [29]. An effect was deemed unclear whether the confidence interval overlapped the thresholds for positive and negative effects. Effect size (ES) magnitudes were calculated as the difference in means/SD for both groups and were qualified as follows: trivial, 0.0–0.2; small, 0.2–0.6; moderate, 0.6–1.2; large, 1.2–2.0; very large, 2.0–4.0; extremely large, 4.0. [30]. ES thresholds for measures of soreness and tiredness were set at 10%, 30%, 50%, 70% and 90% of the VAS range for small, moderate, large, very large and extremely large, respectively [30].

## 3. Results

### 3.1. Within-Group Effects

Post-STG1 analysis of within-group effects revealed decreases in peak torque, CMJ, RSI, sprint performance and RFD, increases in CK and hsCRP, and soreness and tiredness. Minimal effects were seen on oxidative stress markers. The impact of STG2 on muscle function was minimal, with mostly trivial or unclear outcomes, though there was a trivial decrease in sprint performance. STG2 resulted in increased CK, hsCRP for CHO, and LOOHs for MILK. Mean effects, ±90% CI, with qualitative inferences and effect size are presented in Appendix A.

### 3.2. Between-Group Effects

#### 3.2.1. Muscle Function

Baseline (B) peak torque values for knee extension at 60°/s for MILK and CHO were 156.7 ± 27.3 Nm and 163.0 ± 14.9 Nm respectively. There was a benefit for MILK at B-24 h (−5.9 ± 6.8% vs. −10.0 ± 5.8%) and B-48 h (−4.5 ± 8.6% vs. −9.3 ± 7.8%). B-72 h effects were unclear and B-96 h effects were trivial. For 60°/s leg flexion baseline values for MILK and CHO were 81.4 ± 9.3 Nm and 81.7 ± 11.2 Nm. A benefit of MILK was observed at B-24 h (−4.2 ± 13.7% vs. −11.3 ± 7.8%), B-48 h (−4.7 ± 12.4% vs. −14.1 ± 13.5%) and B-72 h (−2.2 ± 13.8% vs. −10.6 ± 15.8%). See Figure 1a.

Baseline values for knee extension at 180°/s were 106.5 ± 19.8 Nm and 113.0 ± 11.9 Nm for MILK and CHO respectively. Outcomes were trivial (B-24 h, B-96 h) and unclear (B-48 h, B-72 h). For knee flexion at 180°/s, baseline values for MILK and CHO were 67.8 ± 11.7 Nm and 66.9 ± 11.3 Nm. A benefit for MILK was seen at B-48 h (−3.0 ± 2.8% vs. −8.9 ± 6.3%). Other comparisons were unclear.

Baseline values for RFD were 429.2 ±125.2 Nm·s^−1^ and 448.1 ± 151.9 Nm·s^−1^. Unclear outcomes were observed B-24 h, B-72 h and B-96 h, with a trivial outcome at B-48 h. Baseline CMJ values for MILK and CHO were 25.8 ± 4.5 cm and 27.7 ± 10.6 cm. Comparisons at all timepoints were unclear.

RSI values at baseline were 105.8 ± 17.2 cm·s^−1^ and 113.6 ± 18.3 cm·s^−1^ for MILK and CHO. Comparisons at B-24 h and B-48 h were unclear. Comparisons from B-72 h (−17.3 ± 10.4% vs. −7.2 ± 24.9%) and B-96 h (−9.5 ± 10.8% vs. −0.3 ± 25.2%) revealed a possible small harmful effect of milk.

Baseline values for 5 m sprint performance were 1.17 ± 0.10 s and 1.18 ± 0.04 s for MILK and CHO. A benefit of MILK was seen at B-24 h (−1.4 ± 3.0% vs. −3.1 ± 6.3%), B-48 h (−1.5 ± 4.1% vs. −3.9 ± 7.0%) and at B-72 h (−2.3 ± 4.4% vs. −5.0 ± 9.5%). Comparisons at B-96 h were unclear. See Figure 1b. Baseline 10 m sprint performance was 2.00 ± 0.10 s and 2.04 ± 0.07 s for MILK and CHO. A trivial outcome was noted at B-24 h, with a benefit of MILK at B-48 h (−1.3 ± 3.1% vs. −2.8 ± 3.5%), B-72 h (−2.0 ± 2.9% vs. −2.6 ± 4.0%) and B-96 h (0.7 ± 2.1% vs. −1.1 ± 2.4%). Baseline values for 20 m sprint were 3.47 ± 0.20 s and 3.55 ± 0.17 s for MILK and CHO. A trivial outcome was observed at B-24 h, with unclear outcomes at other times.

#### 3.2.2. Muscle Soreness and Tiredness

All comparisons of soreness during a squat were unclear. For soreness during isokinetic testing, a possible benefit was seen for MILK at B-24 h (51.0 ± 7.4 vs. 65.5 ± 13.8%). Other comparisons were unclear. For tiredness, a trivial outcome was noted at B–24 h with a benefit for MILK at B-72 h (49.0 ± 17.9 vs. 60.0 ± 19.4%). Other comparisons were unclear.

#### 3.2.3. DALDA

For DALDA B, a benefit of MILK was observed at B-24 h (11.6 ± 8.9% vs. 21.2 ± 11.3%), B-48 h (17.2 ± 14.4% vs. 32.0 ± 11.8%) and B-72 h (16.8 ± 15.5% vs. 29.6 ± 14.6%), with an unclear outcome at B-96 h. See Figure 1c.

#### 3.2.4. Serum Measures

Baseline CK values were 196.2 ± 130.0 and 188.6 ± 125.0 U/L for MILK and CHO respectively. Unclear outcomes were found all timepoints. See Figure 1d. Baseline hsCRP values were 1.6 ± 2.0 mg/L and 1.1 ± 0.6 mg/L for MILK and CHO. A harmful effect was observed for MILK versus CHO at B-2h (1.1 ×/÷ 1.6 vs. 0.9 ×/÷ 1.6), B-24 h (1.5 ×/÷ 1.8 vs. 0.9 ×/÷ 1.8) and B-48 h (1.2 ×/÷ 2.0 vs. 0.7 ×/÷ 1.9). Comparisons at B-72 h and B–96 h were unclear. Baseline LOOH values were 0.6 ± 0.2 μmol/L and 0.6 ± 0.2 μmol/L for MILK and CHO. An unclear outcome was observed at B-2h. A harmful effect was observed for MILK at B-24 h (1.0 ×/÷ 1.2 vs. 0.9 ×/÷ 1.2), with unclear outcomes at all other times. Baseline PC values were 0.34 ± 0.33 nmol/mg protein and 0.27 ± 0.06 nmol/mg protein for MILK and CHO. Unclear outcomes were found at all timepoints.

Mean effects for all measures of muscle function, soreness and tiredness, symptoms of stress, CK, hsCRP, LOOHs and PCs can be seen in Table 1.

### 3.3. Within-Group Effects for STG Variables

Changes in STG variables from STG1 to STG2 were observed. Analysis of within-group effects revealed a decrease in 5 m sprint performance, but a likely increase in 15 m sprint performance for both MILK and CHO. Trivial effects were seen for CMJ, lap time and HR for MILK and CHO. HR was likely lower in STG2 for CHO, but the outcome was trivial for MILK. Changes can be seen in Appendix A.

### 3.4. Between-Group Effects for STG Variables

Comparisons of MILK vs. CHO for 5 m sprint performance (Figure 2a), 15 m sprint performance (Figure 2b), lap times (Figure 2d) and RPE (Figure 2f) were unclear. For CMJ (Figure 2c) a trivial outcome was observed for STG1 vs. STG2 (−0.8 ± 6.8% vs. −0.4 ± 3.3%, MILK vs. CHO). Comparison of HR for MILK vs. CHO (Figure 2e) showed a trivial outcome. Mean effects are shown in Table 2. 

## 4. Discussion

This study is the first to examine the effect of cow’s milk on recovery from repeated simulated team-sport games in females. The principal findings were that cow’s milk had a positive effect on attenuating losses in peak torque, sprint performance and symptoms of stress, and on increases in some measures of muscle soreness and tiredness following STG1. Cow’s milk did not benefit any variable within STG2. STG2 did not lead to a secondary decrease in muscle function (apart from sprint performance, though this was not consistent across groups), soreness, tiredness or perceptions of stress. There was a post-STG2 increase in CK and in hsCRP (CHO) and LOOHs (MILK).

The consumption of cow’s milk had a beneficial effect on muscle function, specifically peak torque and sprint performance (5 and 10 m). Similar benefits have been noted when cow’s milk was consumed following repeated sprinting and jumping in females [12]. This beneficial effect speculatively arises from a post-exercise enhancement of protein balance resulting in a stimulation of protein synthesis rates [31]. For RFD and RSI, comparisons of MILK vs. CHO revealed unclear outcomes with MILK possibly harmful for RSI from B-72 h and B-96 h. There was considerable variability in these variables perhaps due to the more technical nature of the measures. Unclear outcomes were also observed for CMJ. This is in contrast to previous research which reported a beneficial effect of cow’s milk on recovery of CMJ following repeated sprinting and jumping [12]. The effect size for the positive effect of cow’s milk on post-exercise muscle function was small to moderate. The amount of protein provided in the 500 mL of milk was 17 g, giving an intake of 0.26 g/kg. This low overall and relative intake of protein may not have maximally stimulated protein synthesis. It is below the recommended 20 g threshold for maximal stimulation of protein synthesis [32], and substantially lower than the 40 g recently found to be of greater benefit than 20 g following whole-body exercise [33]. Future investigations should be cognizant of this.

The effect of cow’s milk on measures of soreness and tiredness was primarily unclear, though a likely benefit for MILK was noted for tiredness from baseline-72 h. Given that soreness is a subjective measure and that this investigation employed an independent groups design, it is difficult to compare conditions. Cow’s milk had beneficial effects on perceptions of the symptoms of stress, as previously observed [12], with participants who consumed cow’s milk reporting fewer ‘worse than normal’ symptoms over the recovery period. While it is plausible that bias existed because of a lack of blinding to the drink consumed, an attenuation of negative aspects of psychological wellbeing is likely to be positive for athletes.

Comparison of hsCRP data indicates a likely harmful effect of MILK over the 48 h post STG1, though an unclear outcome following STG2. This early benefit for CHO is a result of a decrease in hsCRP levels from baseline to 2 h and baseline to 48 h, while the MILK group showed increases during this time. It has been shown that carbohydrate consumption attenuates increases in inflammatory markers following intermittent exercise [34]. However, this does not explain the lack of benefit for CHO following STG2, and importantly the 500 mL of cow’s milk also contained carbohydrate (25.5 g). Previous research with female soccer players reported an increase in both pro- and anti-inflammatory cytokine responses following a match, with a dampened response following a second game 72 h later [35]. The findings of this study show an opposite effect for CHO, with an increase in hsCRP following STG2. Importantly, hsCRP may not be reflective of the overall cytokine response and research is warranted to investigate this further.

In accordance with previous research with female athletes [11,36], the effect of cow’s milk on the accumulation of CK was unclear. Peak CK was lower than that reported in previous studies utilising variations of the STG [37,38,39]. However, participants in the aforementioned studies were male and it has been reported that oestrogen has a protective effect on the myofibrillar membrane resulting in a smaller increase in CK following exercise [40]. We found no evidence that MILK attenuated oxidative stress as comparisons of MILK vs. CHO for LOOHs and PCs were typically unclear. However, this is consequential to the observation that we did not see clear evidence of oxidative stress throughout the trial. This was unexpected as previous investigations have reported increases in oxidative stress following participation in both repeat-sprint trials [41] and team-sport matches [1]. The contradictory findings in the oxidative stress response between this and previous investigations could be explained by the use of different biochemical markers or analytical techniques. There is also the possibility that oxidative stress did occur but was confined to the muscle tissue and was not detected in the circulation, which is conceivable as the muscle is the prime producer of ROS following exercise [42]. Finally, it is reasonable to speculate that the participants’ basal antioxidant status was more than sufficient to manage any increase in oxidative stress during and after exercise.

Interestingly, despite cow’s milk benefiting muscle function measures over the recovery period, a comparison of sprint performance and CMJ from STG1 to STG2 revealed no clear benefit of MILK. Of relevance, Mohr et al. [8] highlighted no difference in peak sprinting speed and sprinting distance when 3 simulated games were played within 7 days, despite increases in other markers of fatigue and muscle damage. Of note in the current study is that the average 5 m sprint time in STG1 and STG2 (~1.37 s) was considerably slower than the isolated sprint times measured during recovery (~1.20 s), and CMJ height was approximately 80% of the isolated jump height, indicating that participants did not sprint nor jump maximally during the STG. This is most likely related to the physiological demands of the STG, but nonetheless questions the validity of using isolated sprint and jump performance as a marker of recovery, or a predictor of performance for subsequent match situations. Furthermore, based on the results of this study, it is conceivable that the positive effect of cow’s milk is greatest when participants are maximally performing. What we understand from this is that isolated measures of sprint performance and jump height are not indicators of the performance of these measures within a simulated team-sport situation. Thus, crucially, future research should consider the validity of indicators of recovery.

Surprisingly, STG2 had little impact on measures of muscle function, soreness, tiredness or perceptions of stress. Differences from 48–72 h and 48–96 h were typically trivial or unclear with some measures (peak torque, CMJ, RSI, soreness, tiredness, DALDA) continuing to return towards baseline values despite the demands of STG2, the exception being sprint performance. Nonetheless, performance of all muscle function variables remained below baseline at 96 h, which has implications for coaches and athletes in terms of training programmes and competition fixtures. The two-fold increase in the pattern of CK was as expected, with an increase observed following STG1, returning towards baseline before a second increase following STG2. This is similar to the pattern observed following repeated match play [3,8], though this pattern is clearly not aligned to that of muscle function. The other serum markers were either not impacted by STG2 (PCs) or not impacted across both groups (LOOHs, hsCRP), reflecting variability in the measures.

Limitations of this study include the small sample size and the use of an independent groups design. The latter, however, was chosen to reduce the impact of a Repeated Bout Effect whereby participation in an exercise bout with a high eccentric load results in reduced muscle damage in the next subsequent bout of exercise [43]. Despite the attempt to use an exercise protocol that simulates the physiological demands of team sports, there are some non-physiological factors missing from the protocol that could significantly influence performance in real-game situations. These include contextual factors such as match status/score, lack of an opponent, an unrealistic match location and lack of physical contact. Importantly, the protocol did not include kicking, catching or striking activities which would not only influence physiological factors during the protocol, but possibly the level of post-exercise fatigue and muscle damage.

## 5. Conclusions

In conclusion, the present study clearly demonstrates that the consumption of cow’s milk following two simulated team games separated by 48 h can attenuate losses in peak torque, sprint performance and symptoms of stress compared to an energy-matched carbohydrate drink in female athletes, with benefits also for some measures of soreness and tiredness. From a practical perspective, this study indicates that cow’s milk may serve to enhance the recovery of female athletes following team-sport participation. No benefit of cow’s milk was observed for sprint performance, jump height or lap times within the second STG protocol, which questions how transferable isolated measures of recovery are to real match performances. Further research is necessary to elucidate whether cow’s milk can affect within-game performance variables when recovery periods are limited.

## Figures and Tables

**Figure 1 nutrients-12-00112-f001:**
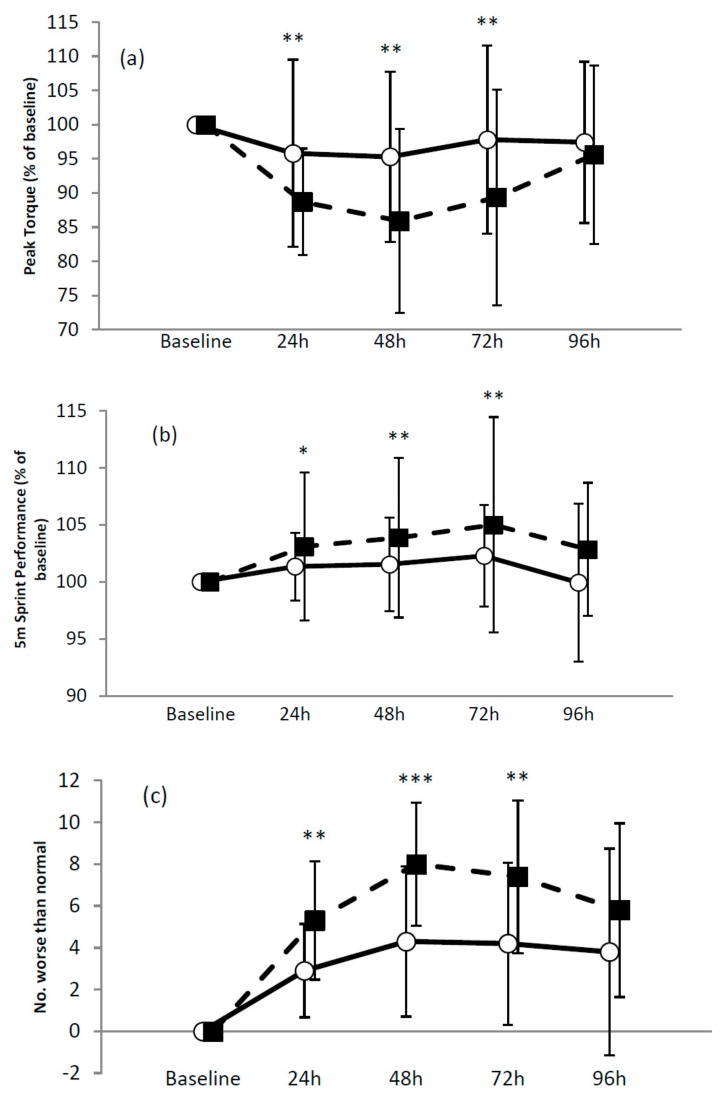
Effect of milk vs. carbohydrate on recovery from repeated simulated team-sport games. (**a**) Peak torque 60°/s for knee flexion; (**b**) 5 m sprint performance; (**c**) Daily Analysis of Life Demands (DALDA) Part B; (**d**) Creatine kinase. Values are presented as the mean ± SD. ◯ Cow’s milk (MILK); ⬛ an energy-matched carbohydrate (CHO). * Possible; ** likely; *** very likely.

**Figure 2 nutrients-12-00112-f002:**
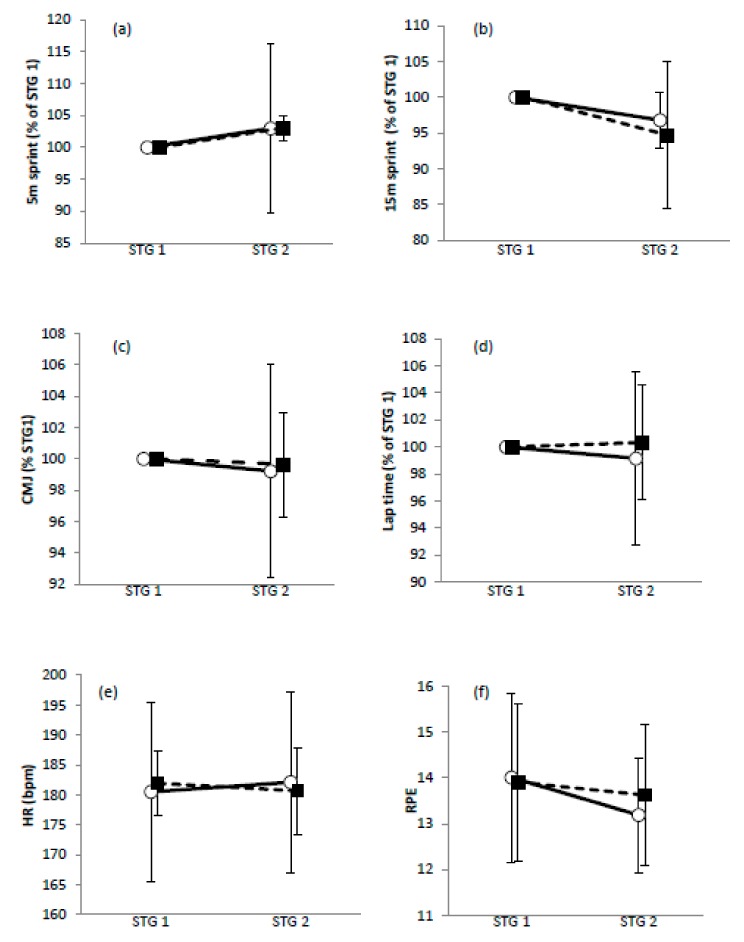
Mean values for simulated team game (STG) 1 vs. STG 2 (**a**) 5 m sprint performance; (**b**) 15 m sprint performance; (**c**) countermovement jump performance; (**d**) lap time; (**e**) heart rate; (**f**) rate of perceived exertion. Values are presented as the mean ± SD. ◯ MILK; ⬛ CHO.

**Table 1 nutrients-12-00112-t001:** Effects of MILK vs. CHO on muscle function, soreness, tiredness, symptoms of stress, serum creatine kinase (CK), high sensitivity C-reactive protein (hsCRP), LOOHs and PCs following repeated simulated team-sport game.

Variable	Time Frame	Mean Effect ^a^, ± or ×/÷ 90% CI ^b^	Qualitative Inference ^c^
			(ES)
Peak Torque 60°/s Leg extension	B-24 h	5.1, ±5.2	Small (0.26) benefit **
	B-48 h	5.2, ±6.7	Small (0.28) benefit *
	B-72 h	2.5, ±6.5	Unclear
	B-96 h	0.3, ±5.6	Trivial *
Peak Torque 60°/s Leg flexion	B-24 h	7.4, ±8.6	Small (0.45) benefit **
	B-48 h	11.4, ±13.6	Moderate (0.61) benefit **
	B-72 h	10.5, ±15.2	Small (0.52) benefit **
	B-96 h	2.3, ±11.5	Unclear
Peak Torque 180°/s Leg extension	B-24 h	−1.7, ±5.4	Trivial *
	B-48 h	1.5, ±6.9	Unclear
	B-72 h	3.0, ±7.1	Unclear
	B-96 h	−0.9, ±6.6	Trivial *
Peak Torque 180°/s Leg flexion	B-24 h	2.7, ±7.4	Unclear
	B-48 h	6.6, ±4.6	Small (0.32) benefit **
	B-72 h	2.2, ±8.8	Unclear
	B-96 h	0.9, ±9.4	Unclear
Rate of Force Development (RFD) (0–200 ms)	B-24 h	18.8, ±46.6	Unclear
	B-48 h	2.2, ±28.0	Trivial *
	B-72 h	1.0, ±35.8	Unclear
	B-96 h	23.9, ±45.6	Unclear
Countermovement Jump (CMJ) height	B-24 h	0.4, ±6.8	Unclear
	B-48 h	2.9, ±9.2	Unclear
	B-72 h	1.5, ±6.8	Unclear
	B-96 h	1.5, ±8.4	Unclear
Reactive Strength Index (RSI)	B-24 h	7.4, ±11.1	Unclear
	B-48 h	1.1, ±10.8	Unclear
	B-72 h	−8.4, ±15.9	Small (0.47) harm *
	B-96 h	−7.8, ±14.3	Small (0.48) harm *
5 m sprint	B-24 h	−1.9, ±2.4	Small (0.40) benefit *
	B-48 h	−2.5, ±3.2	Small (0.51) benefit **
	B-72 h	−2.7, ±3.6	Moderate (0.58) benefit **
	B-96 h	−3.2, ±5.0	Unclear
10 m sprint	B-24 h	−0.8, ±2.3	Trivial **
	B-48 h	−2.1, ±2.6	Small (0.50) benefit **
	B-72 h	−1.3, ±2.6	Small (0.30) benefit **
	B-96 h	−2.3, ±1.6	Small (0.53) benefit *
20 m sprint	B-24 h	0.1, ±2.1	Trivial *
	B-48 h	−0.3, ±2.5	Unclear
	B-72 h	−0.1, ±2.0	Trivial *
	B-96 h	−0.1, ±3.5	Unclear
Muscle soreness (Squat)	B-24 h	−0.2, ±1.7	Unclear
	B-48 h	−1.1, ±1.3	Unclear
	B-72 h	−0.6, ±1.6	Unclear
	B-96 h	−0.5, ±1.5	Unclear
Muscle soreness (Isokinetic knee extension/flexion)	B-24 h	−0.7, ±0.8	Trivial benefit *
	B-48 h	−0.1, ±1.1	Unclear
	B-72 h	−0.5, ±1.1	Unclear
	B-96 h	−0.8, ±1.5	Unclear
Muscle tiredness	B-24 h	−0.1, ±1.4	Unclear
	B-48 h	−1.0, ±1.5	Unclear
	B-72 h	−1.1, ±1.5	Small benefit **
	B-96 h	−0.5, ±1.4	Unclear
DALDA Part B	B-24 h	−2.4, ±2.0	Small benefit **
	B-48 h	−3.7, ±2.6	Small benefit ***
	B-72 h	−3.2, ±2.9	Small benefit **
	B-96 h	−2.0, ±3.6	Unclear
Creatine Kinase	B-2 h	1.0, ×/÷1.5	Unclear
	B-24 h	1.0, ×/÷1.8	Unclear
	B-48 h	0.9, ×/÷ 1.5	Unclear
	B-52 h	1.1, ×/÷1.6	Unclear
	B-72 h	1.0, ×/÷1.9	Unclear
	B-96 h	1.0, ×/÷1.5	Unclear
hsCRP	B-2 h	1.3, ×/÷1.5	Small (0.48) harm **
	B-24 h	1.7, ×/÷1.6	Moderate (0.76) harm **
	B-48 h	1.7, ×/÷1.7	Moderate (0.61) harm **
	B-52 h	1.0, ×/÷2.1	Unclear
	B-72 h	1.0, ×/÷2.2	Unclear
	B-96 h	1.0, ×/÷2.1	Unclear
Lipid Hydroperoxides (LOOHs)	B-2 h	1.2, ×/÷1.5	Unclear
	B-24 h	0.9, ×/÷1.2	Small (0.18) harm **
	B-48 h	1.1, ×/÷1.3	Unclear
	B-52 h	1.4, ×/÷1.7	Unclear
	B-72 h	1.1, ×/÷1.4	Unclear
	B-96 h	0.9, ×/÷1.4	Unclear
Protein Carbonyls (PCs)	B-2 h	0.9, ×/÷2.0	Unclear
	B-24 h	0.9, ×/÷1.5	Unclear
	B-48 h	0.8, ×/÷1.6	Unclear
	B-52 h	0.8, ×/÷1.5	Unclear
	B-72 h	0.9, ×/÷1.3	Unclear
	B-96 h	0.7, ×/÷1.6	Unclear

^a^ Mean effect refers to MILK minus CHO; ^b^ ±90% CI: add and subtract this number to the mean effect to obtain the 90% confidence intervals for the true difference; **^c^** qualitative Inference represents the likelihood that the true value will have the observed magnitude; * possible; ** likely; *** very likely.

**Table 2 nutrients-12-00112-t002:** Effect of MILK vs. CHO on simulated team-sport game variables, 5 m sprint, 15 m sprint, lap time, jump height, heart rate (HR) and rating of perceived exertion (RPE).

Variable	Mean Effect ^a^ ± 90% CI ^b^	Qualitative Inference ^c^
5 m sprint	−1.5 ± 7.4	Unclear
15 m sprint	−0.8 ± 3.2	Unclear
CMJ	−0.6 ± 4.4	Trivial *
Lap time	−1.1 ± 3.9	Unclear
HR	−0.3 ± 1.3	Trivial **
RPE	−0.6 ± 1.0	Unclear

^a^ Mean effect refers to MILK minus CHO; ^b^ ±90% CI: add and subtract this number to the mean effect to obtain the 90% confidence intervals for the true difference; **^c^** qualitative Inference represents the likelihood that the true value will have the observed magnitude; * possible; ** likely.

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
