# Peer review of "Can Milk Affect Recovery from Simulated Team-Sport Match Play?"

_nutrients, 2019, doi:10.3390/nu12010112_

Round 1

Reviewer 1 Report

In the current study, Rankin and colleagues sought to determine whether milk ingestion following a simulated team game play can improve markers of recovery and enhance subsequent team game play compared to carbohydrate ingestion. The authors employ an parallel group design where each group received either milk or carbohydrate prior to two simulated team game scenarios. The authors found that milk ingestion subtly preserved peak torque of the knee extensors and flexors, 5m sprint performance, and symptoms of stress following a simulated team game versus carbohydrate ingestion alone. However, the authors also state that all measures of performance during the subsequent simulated team game were unaffected by milk.

The manuscript is very well-written and the results are concisely discussed. This work is novel in that it is the first to assess the efficacy of milk as a recovery beverage from simulated team games in female athletes. Below are several minor revisions:

Introduction

- There could be more of an emphasis placed in why we would believe that the physiological response to milk ingestion following STG would differ between males and females.

Methodology

- Why were treatment beverages provided prior to STG?

- Were STG1 and STG2 work-matched?

- Please define all acronyms at the time of first use. Many of the measurements are only referred to by their acronyms.

- The rationale for measuring blood markers of oxidative stress is unclear.

- Was any familiarization of measurement protocols included prior to the first STG?

- Please include a statement justifying your choice of inferential statistics as opposed to traditional hypothesis testing (i.e. ANOVA, T-tests).

Results and Discussion

- The discussion of oxidative stress does not fit with the hypothesis or other measurements. Consider removing.

- The authors need to discuss how nutrient timing can impact recovery from exercise and incorporate the current nutrient timing (prior to exercise) in this discussion.

Reviewer 2 Report

The article examines milk's effect on damage markers after STG. This is a novel approach and designed well. Below are my comments for improvements on the paper. 

Title: I feel that intros should mention the results of the study. 

Abstract and Keywords: No suggestions. Put together nicely. 

Intro: 
Line 37: Comma after In reality, 
Line 41-43: Based on what literature? This needs to be identified and have references
Line 44: Minimising >> minimizing
Line 51-55: Split the sentence (run-on)
Line 56-57: It reads awkwardly, suggest listing two objectives.  

Methods:
Line 65: Need an "and/or" after pregnancy 
Line 66: Comma after "briefings"
May have been a stronger study to have a cross-over design
Overall methods are clear and easy to replicate
Stats: In the sports and exercise world we are going away from the magnitude-based inference stats. The review in MSSE showcases the limitations and the poor inferences. See Welsh, Knight. Magnitude-based Inference: A Statistical Review. MSSE. 2015.47(4): 874-884. It states that authors should be more realistic with their study limitations and use CI or a fully Bayesian analysis. 

Results:

I did not have access to the ES as a supplement. Suggest mentioning them in the manuscript and not as a supplement. 

Based on the stats run the results section is well put together. The additions of the tables and figures complemented the text. Tables and figures were labeled nicely. Figure one does not have a legend to know which line is MILK or CHO

Discussion: 

Discussion is well written and straight to the point. Would like to see more limitations as sample size and design (cross-over) is a limitation. Included a conclusion/practical application. 

Reviewer 3 Report

The tittle of this manuscript is “Can milk affect recovery from simulated team sport match play?” It appears the purpose of this manuscript is to evaluate the impact of cow’s milk consumption (vs. CHO) has on exercise performance and the overall impact it has on muscle function recovery and blood markers. In fact, the authors concluded that cow’s milk consumption can attenuate losses in peak torque, sprints performance and symptoms of stress compared to a CHO beverage consumption. Despite the fact that milk was offered in a fixed quantity (regardless the body weight of the participants), the amount of protein ingested (milk group) can influence the degree of recovery and contribution to muscle protein synthesis (i.e. recovery function), maybe, because the high impact that leucine (present in milk) has on MPS and overall muscle metabolism.

General comments:

It is very important to state cow's milk throughout the entire manuscript. As plant based beverages have surfaced and many consumers refer to for example almond milk as "milk". I’m my opinion it is important to state that cow's milk was used (through the entire manuscript) to avoid any confusion and to clearly define content.

I suggest to the authors to change “milk” and “recovery” from the keywords as both appears in the title and abstract of the manuscript. Maybe “performance” and/or “post-exercise recovery” could be better instead of the aforementioned keywords.

I recommend to the authors to define abbreviations in parentheses the first time they appear in the manuscript, and also in figures and tables to make easier the reading of the manuscript for non-expert readers. Examples:

Line 90: CMJ, HR, and RPE those abbreviations have not been defined previously. Lines 110-116: CK, hsCRP, LOOH and PC. Lines 119-120: VAS Line 137: RSI And also in figures and tables.

Lastly, I recommend to the authors to review the text of the manuscript since there are some errata. Examples:

Line 73, 88, 102, 164… two spaces (please delete 1) Line 128: insert a space between “and” and “180º/s” Line 213: remove “**” from the sentence. Line 238 “lap time” instead of “laptime”

Introduction section:

Lines 51-55: I partially agree with the authors’ statement, however, while some authors and research have clearly suggested that milk could contribute post-exercise recovery (e.g. PMID: 30379113) others did not because the limited number of well-designed studies (controlling the blinding of the beverages, generating a random sequence of beverage groups, etc.) (e.g. PMID: 31060583). I suggest to the authors to mention both general opinions in this paragraph to better describe the importance of their research and the “gap” in the current literature.

Material and Methods section:

Participants subsection:

Why were not presented the participants’ characteristics? I’m my opinion, in a study comparing two “treatments” (MILK vs. CHO) on muscle recovery it is important to know if differences on baseline conditions (e.g. training status) were observed before the treatment and/or study protocol. Were observed differences between groups at baseline? (i.e. after the randomization). Please state if differences were observed (or not) in the results section, as those differences (if exists) could be driving the results. Were the participants of the study professional athletes? Semi-professionals? Amateurs? This information could be relevant to understand the recovery from the exercise protocol (after the beverage consumption).

Exercise protocol subsection:

Camogie and Gaelic football are sports in which different exercise patterns are observed during match-play (e.g. jogging, walking, cruising, sprinting…), however, in the current version of the manuscript is not well explained how the exercise protocol has been performed. Were the intermittent running exercises performed at the same intensities and in the same order? I’m my opinion, as well as the intensity, it is also important to state the order of the intensities as those sports (as well as soccer) has a non-cyclical nature (please see PMID: 11144865). Furthermore, if a systematic intensity order was followed and the non-cyclical nature of the sport did not exist, I’m my opinion, the authors should not refer to the exercise as “simulated team sport match play”, but as intermittent (or cyclic) aerobic exercise for example.

Nutritional intervention and dietary control subsection:

Were the beverages blinded to the participants and/or to the researchers? Please state this information in the methods section as is an important methodological issue. In fact, in a recently published systematic review (effect of milk consumption on exercise performance and recovery), they recommended to improve the study designs to demonstrate if cow’s milk is useful (or not) as a sport nutrition-related supplement (PMID: 31060583).

Blood sampling and biochemical analysis subsection:

Were all the analysis duplicated? If yes, please state the coefficients of variation (CV), if not please state that samples were analysed just once in both, methods and limitation paragraph.

Results section:

Within-group effects subsection:

As I previously mentioned, please, state if differences between groups were observed after the randomization (and before the treatment).

Figure 1 (page 5) and 2 (page 9):

Please differentiate which group represent the circles, and which group represent the squares. Are results presented as means and standard deviation? If yes, please state in the figure explanation to make easier the understanding of the figures. Lastly, in the Figure 1 explanation the concepts are separate by semicolons while in Figure 2 by dots, please be consistent. Figure 1: are the authors comparing between MILK and CHO beverages? Because the Figure 1 explanation is “effect of milk on recovery…”, however, open circles and squares (i.e. 2 groups) are represented, please clarify this fact in the explanation. Please add in the explanation “knee extension and flexion” as was stated in the methods section. Please add to Panel C “DALDA Part B”. Please add Creatine Kinase (CK) in the explanation, and review both figures to complete information (i.e. correct explanation and use of the abbreviations).

Table 1 (pages 6-8):

Please add “game” after “repeated simulated team-sport”. Please add “part” to “DALDA B”. Please add the definition of abbreviations to make easier the reading of the table for non-expert readers.

Table 2 (pages 9-10):

Please add “following repeated simulated team-sport game” to be consistent. In line with the previous comment, in Table 2 authors mentioned “MILK v CHO” while in Table 1 did not. And lastly, please add the definition of abbreviations in both tables.

Discussion section:

A possible limitation of the current study is that MILK was ingested in a fixed quantity (500mL) and was not individualized to participants’ body weight for example. One could hypothesize that the controversial findings on the present study (and well discussed by the authors) could be in a greater or lesser extent explained by this fact. In fact, have been previously recommended the ingestion of 0.8-1.2 g of CHO/kg/h and 0.2-0.4 g protein/kg/h (preferably after exercising), and importantly, with a minimum of 20 g high-quality protein (specially leucine; PMID: 20368372) for improve the post-exercise recovery (PMID: 24180469), however, some controversy still exist regarding the timing and if both, CHO and protein have to be consumed after the exercise (PMID: 24180469 and PMID: 22344059). So, following the aforementioned recommendations maybe the negative (and/or controversial) results could be (partially) explained by an insufficient amount of milk, and thus, for an insufficient amount of protein and/or leucine (i.e. not enough milk). I suggest to the authors to discuss this fact in the discussion section as well as to mention in the limitation paragraph the fact of the fixed quantity of milk (i.e. of protein) regardless the body weight of the participants. Maybe, this could be also a new line for future studies that should be acknowledged in the future lines section/discussion section, I mean to individualize the amount of milk ingested (to individualize the amount of protein/leucine) and to study the effect of the individualized ingestion on post-exercise recovery.

Supplementary material:

Were the results presented from DALDA Part A or B? I suppose that are from part A as B are presented in the main manuscript, but please, clarify this.

Round 2

Reviewer 3 Report

The manuscript is much improved and provides valuable additional information.

Only two minor comments:

Line 145: the term "Countermovement jump" was abbreviated in the "Exercise Protocol" section, so "CMJ" can be used directly. Figure 1 and 2 (footnotes): group identification (i.e. squares) is partially covering "CHO"

Author Response

Responses to Reviewer 3

Line 145: the term "Countermovement jump" was abbreviated in the "Exercise Protocol" section, so "CMJ" can be used directly.

Amended

Figure 1 and 2 (footnotes): group identification (i.e. squares) is partially covering "CHO"

Figures saved as both word and pdf documents with correction